# False versus True Statin Intolerance in Patients with Peripheral Artery Disease

**DOI:** 10.3390/jcm11226619

**Published:** 2022-11-08

**Authors:** Jörn F. Dopheide, Patrick Gillmann, David Spirk, Meisam Khorrami Borozadi, Luise Adam, Heinz Drexel

**Affiliations:** 1Clinic for Angiology, Swiss Cardiovascular Center, Inselspital, Bern University Hospital, University of Bern, 3010 Bern, Switzerland; 2Department of Medicine I, Division of Cardiology, Angiology and Intensive Medical Care, University Hospital Jena, Friedrich-Schiller-University Jena, 07743 Jena, Germany; 3Vorarlberg Institute for Vascular Investigation and Treatment (VIVIT), 6800 Feldkirch, Austria; 4Institute of Pharmacology, University of Bern, 3010 Bern, Switzerland; 5Institute of Primary Health Care (BIHAM), University of Bern, 3010 Bern, Switzerland; 6Department of General Internal Medicine, Inselspital, Bern University Hospital, University of Bern, 3010 Bern, Switzerland; 7Private University of the Principality of Liechtenstein, 9495 Triesen, Liechtenstein; 8Drexel University College of Medicine, Philadelphia, PA 19129, USA; 9Department of Internal Medicine, Academic Teaching Hospital Bregenz, 6900 Bregenz, Austria

**Keywords:** lipid-lowering therapy, low-density lipoprotein cholesterol, mortality, peripheral artery disease, statin intolerance

## Abstract

Background: Statin intolerance (SI) is often documented in patients’ charts but rarely confirmed by objective methods. Objective: We aimed to identify the rate of true SI in a large population with peripheral artery disease (PAD) as well as the subsequent use of such drugs and the impact on cardiovascular outcomes. Methods: Patients with PAD and reported SI were retrospectively classified in those with “probable/possible” (pp) and “unlikely” (u) SI, after the application of the “Statin Myalgia Clinical Index Score” (SAMS-CI). Both groups were compared after 62 months (date of observation period?). Results: Among the 4,505 included patients, 139 (3%) had been reported as having SI. Of those, 33 (24%) had ppSI, and 106 (76%) had uSI. During the observation period, statin use decreased in patients with both ppSI (from 97% to 21%; *p* < 0.0001) and uSI (from 87% to 53%; *p* < 0.0001). At the end of the observation period, patients with ppSI more often received PCSK9 inhibitors (55% vs. 7%; *p* < 0.0001), had a stronger decrease in LDL-C from baseline to follow-up (1.82 ± 1.69 mmol/L vs. 0.85 ± 1.41 mmol/L; *p* < 0.01), and a lower rate of mortality (3% vs. 21%; *p* = 0.04) than those with uSI. Conclusions: SI is low in PAD patients (3.1%), with only one quarter fulfilling the criteria of ppSI. The overdiagnosis of SI is related to an underuse of statins and an increased mortality in a short time period.

## 1. Introduction

At present, cardiovascular disease still has the highest death toll worldwide. PAD patients suffer from a particularly poor prognosis. With this knowledge, efforts to reduce ischemic events and improve cardiovascular outcomes have been the driving force behind cardiovascular research in recent decades.

One central strategy to improve cardiovascular outcomes is lipid lowering. For the time being, the most important drugs to reduce cholesterol are still statins. They are widely known to be beneficial in both primary and secondary prevention of major cardiovascular events (MACE) [1,2], and there is convincing evidence that they consistently reduce total and cardiovascular mortality [3,4,5,6].

Despite these proven beneficial effects, adherence to guideline-recommended statin therapy is suboptimal [7,8,9], although the vast majority of patients tolerate statins well. Nevertheless, some patients do not, and statin-associated or -non-associated adverse symptoms likely contribute significantly to the high discontinuation rates of statin therapy (up to 75%) within two years of therapy initiation [10]. Such treatment non-adherence/discontinuation significantly impacts the occurrence of MACE as well as total and cardiovascular mortality [11,12].

Data on statin adherence and statin intolerance (SI) in peripheral artery disease (PAD) are scarce, thus the estimated rates of SI may be incorrect [9]. Furthermore, the clinical presentation of potential statin-related symptoms is highly heterogeneous, making the correct diagnosis of SI difficult and demanding. Specifically, PAD patients might misinterpret ischemic muscle pain to be statin-induced and vice versa. Thus, some patients might be tagged with a false diagnosis of SI exposing them to an unnecessarily higher cardiovascular risk due to inadequate lipid-lowering treatment (LLT). Distinguishing statin-induced from non-related myopathy and treating the right patients with optimal LLT is therefore of utmost importance to further reduce the burden of cardiovascular disease including mortality.

In the present study, we aimed to investigate the frequency of documented SI, true or false, in consecutive patients with PAD and evaluate the LLT patterns and clinical outcomes in these patients.

## 2. Patients and Methods

### 2.1. Patients and Study Design

This is a retrospective study from a single-center registry. Overall, 4505 consecutive patients were referred for asymptomatic or symptomatic PAD (Fontaine stage I-IV) to the outpatient clinic of our tertiary reference center from January 2009 until December 2019 for diagnosis and treatment. The local ethics committee approved the use of patient data for this registry (Ethics approval ID: 2019-00357).

#### 2.1.1. Inclusion Criteria

Patients were included if statin intolerance was documented in their hospital database charts while treated for PAD (Fontaine stage I–IV), were older than 40 years, and had a baseline lipid profile.

#### 2.1.2. Exclusion Criteria

Patients without formal consent were excluded as were those with triglycerides (TG) >4.5 mmol/L in order to be able to calculate low-density lipoprotein cholesterol (LDL-C) according to the Friedewald formula [13]. Non-atherosclerotic vascular diseases Furthermore, patients with a concomitant diagnosis of cancer, autoimmune disease, or chronic infectious disease (i.e., HIV) or transplantation requiring immunosuppressive medication, as well as any other medication potentially interfering with statin therapy, were excluded to avoid situations potentially covering a true statin intolerance.

### 2.2. Definition of Statin Treatment

In terms of intensity, statin therapy at baseline was classified as high, moderate, or low. High intensity was defined as treatment with Atorvastatin 40–80 mg or Rosuvastatin 20–40 mg, moderate intensity was Atorvastatin 10–20 mg, Fluvastatin 40 mg, Pravastatin 40–80 mg, Rosuvastatin 5–10 mg, Simvastatin 20–40 mg, or Pitavastatin 2–4 mg, and low intensity was Fluvastatin 20–40 mg, Pravastatin 10–20 mg, Simvastatin 10 mg, or Pitavastatin 1 mg [14,15]. The mean prescribed statin dosage was normalized to Simvastatin 40 mg [15].

LDL-C goals were assessed according to the 2016 and 2019 ESC/EAS guidelines on the management of dyslipidemias, first with LDL-C <1.8 mmol/L [3] and second with <1.4 mmol/L [16], respectively, in order to account for the recent goal evolutions in the latest guidelines.

### 2.3. Evaluation of Statin Intolerance

Statin intolerance was defined, according to a consensus paper of the European Atherosclerosis Society (EAS), as muscular complaints caused by at least two different statins having occurred within 4 to 6 weeks of treatment initiation [17]. The muscle complaints had to be typically symmetrical and proximal, affecting large muscle groups, including the thighs, buttocks, calves, or back muscles [17]. To evaluate the probability of potential statin intolerance, we used the “statin myalgia clinical index score” [18] based on the findings of the “**ST**atins **O**n **M**uscle **P**erformance (STOMP)” trial [19]. According to this score, all enrolled patients were classified as having unlikely or possible/probable statin intolerance for the purpose of our analysis.

### 2.4. Follow-Up in the Lipid Outpatient Clinic and Definition of Clinical Outcomes

In 2016, the Division of Angiology at the Bern University Hospital instituted an outpatient lipid clinic, implemented within the regular angiology outpatient service. This clinic was set up for all referred PAD patients with LDL-C levels above 1.8 mmol/L (and as of 2019 above 1.4 mmol/L) for intensification of their LLT. Patients with potential statin intolerance were also seen in the lipid clinic for further evaluation of their statin medication and any potential side effects. LLT modifications were instituted and monitored until an optimal and effective LLT regimen was established.

For endpoint assessment, patient’s charts from the general hospitals patient data system were screened for death and major adverse cardiovascular events (MACE) as assessed by the treating physicians. The primary endpoint was time to all-cause death defined as mortality from any reason including cardiovascular death. Secondary endpoints were cardiovascular (CV) death defined as mortality from any cardiovascular death, and cardiovascular event (CVE), defined as stroke, myocardial infarction (ST- or non-ST- elevation), or major adverse limb events (MALE), the latter being defined as amputation above the ankle or revascularization procedure

### 2.5. Statistical Analysis

Categorical data are presented as absolute numbers and percentages (%). Continuous variables are described as means with standard deviations (±SD). Subgroups of categorical data were compared using Fisher’s exact test. For comparisons of two groups with continuous variables, we used the Mann–Whitney–Wilcoxon test, and for three-group comparisons, we used the Kruskal–Wallis test. Two-tailed *p*-values < 0.05 were considered significant.

All statistical analyses were performed using the GraphPad Prism statistical software package, version 8.4.3 (GraphPad Software, San Diego, CA, USA).

## 3. Results

### 3.1. Patients Characteristics and Identification of Patients with Possible or Probable Statin Intolerance

From 4505 PAD patients treated during the observational period, we identified 139 PAD patients with a diagnosis of statin intolerance or statin myopathy documented in their medical records and enrolled them in our study. Patients falsely diagnosed with PAD were excluded from the study, although potentially having SI. In these cases, arterial occlusion was of embolic origin (i.e., mostly cardiac) or due to non atherosclerotic vessel wall alterations, (i.e., popliteal entrapment, vasculitis, etc.).

We identified 106 (76%) with unlikely, 6 (4%) with possible, and 27 (20%) with probable statin intolerance (SI) (*p* < 0.0001), according to the “statin myalgia clinical index score” [18]. Putting this in relation to the total number of PAD patients treated in our facility during the observational period, even lower proportions for unlikely, probable, and possible SI were present (Figure 1).

Baseline characteristics according to the probability of SI are presented in Table 1. Patients with uSI were older and had higher creatinine values than patients with a ppSI, whereas LDL-C or non-high-density lipoprotein cholesterol (non-HDL-C) were higher in ppSI patients.

### 3.2. Statin Therapy and Lipid Profile at Baseline

The baseline data on statin use reflect the prescribing habits of external providers (GPs). At baseline, 82% of uSI and 97% of ppSI (*p* = 0.04) patients had been treated with statins with a mean statin dose of 57.8 ± 34.9 mg, normalized to Simvastatin. Muscle symptoms during statin intake were reported in total in 89 (64%) patients, but typical muscle symptoms were mainly reported in the ppSI group (*p* < 0.0001). uSI patients (78%) mainly reported atypical, rather unspecific symptoms, such as a gastrointestinal disorder or unspecific sensations related to statin use.

The majority of ppSI patients tried more than one statin (88%), compared to uSI patients of which only 47% tried more than one statin (*p* < 0.0001). At baseline, the majority of patients in both groups had been prescribed a high-intensity treatment with atorvastatin (76%) or rosuvastatin (88%) at least once since initiation. Furthermore, ezetimibe was used solely in combination with statins. Baseline statin-related data according to the probability of statin intolerance are presented in Table 2 and Appendix A.

### 3.3. Lipid-Lowering Therapy over Time

After the evaluation of the likelihood of a ppSI, we analyzed the LLT regimen installed after the first referral to our institution. During the observational period, 76 patients (55%) were reluctant to take any further statin. Of these, 50 (36%) patients were identified with an uSI, versus 26 (19%) patients having a ppSI.

Statin therapy was then prescribed in 63 (45%) patients, in combination with ezetimibe (*n* = 18) or in combination with a PCSK-9 inhibitor (2 Alirocumab, 3 Evolocumab). Apart from this, PCSK-9 inhibitors were prescribed in a further 20 cases (11 Alirocumab, 9 Evolocumab), with Ezetimibe given in 3 cases as an additional lipid-lowering agent. Table 2 summarizes the LLT at baseline and follow-up for all patients, as well as for unlikely or probable/possible SI separately.

Statin intensity changed from high (24% decrease, *p* = 0.003) to moderate (22% increase, *p* = 0.005), still with Atorvastatin (*n* = 9) and Rosuvastatin (*n* = 22) being used in the majority, but with moderate dosages (mean Simvastatin equivalent 51 mg/day). Especially in ppSI patients, the majority (86%) received a moderate statin regimen with atorvastatin (57%) or rosuvastatin (29%) in tolerable dosages (mean Simvastatin equivalent to 37 mg/day).

### 3.4. LDL-C Levels over Time

Mean LDL-C levels at baseline were 2.79 ± 1.18 mmol/L. The majority of patients (79%) were above 1.8 mmol/L, the recommended target level at that time (Figure 2a–f). During the follow-up, mean LDL-C levels decreased from 2.79 ± 1.18 mmol/L to 2.11 ± 1.29 mmol/L; *p* < 0.0001). In patients with unlikely statin intolerance, mean LDL-C levels decreased from 2.64 ± 1.24 mmol/L to 2.24 ± 1.32 mmol/L (*p* = 0.25), whereas in patients with possible/probable statin intolerance, mean LDL-C levels decreased from 3.25 ± 1.14 mmol/L to 1.78 ± 1.19 mmol/L, *p* < 0.0001 (Figure 3a). Furthermore, the mean individual reduction of LDL-C showed a higher decrease in patients with possible/probable versus unlikely statin intolerance (−1.82 ± 1.69 mmol/L vs. −0.85 ± 1.41 mmol/L, respectively; *p* < 0.01; Figure 3b).

### 3.5. Clinical Endpoints

The mean follow-up time was 62 ± 32 months (minimum of 2 and maximum of 133 months). Overall, 23 (17%) patients died, of which 7 (30%) were considered CV deaths. MACE occurred in 107 (77%) patients, with a total of 315 events documented during the follow-up period.

All-cause mortality was higher in patients with uSI versus ppSI (uSI *n*= 22 vs. ppSI *n* = 1; 95% CI 2.356 to 16.21, *p* = 0.04; Figure 4). The rate of CV deaths and CVE was similar between patients with uSI versus ppSI (CVD: uSI *n* = 7 vs. ppSI *n* = 0; *p* = 0.21, 95% CI 0.03805 to 2.042); (CVE: uSI *n* = 69 vs. ppSI *n* = 24, *p* = 0.11; 95% CI 0.4155 to 1.160) (see Appendix A).

## 4. Discussion

From the present observational study of 4505 PAD patients, we report four major findings: (i) The majority of PAD patients (three out of four) with documented statin intolerance (*n* = 139) are rather unlikely to suffer from this condition, further indicating that (ii) the potential frequency of likely statin intolerance in the PAD population might be lower than anticipated (<1%), (iii) an erroneously assumed statin intolerance might lead to a suboptimal if non-existent LLT, thus potentially leads to (iv) higher overall mortality in PAD patients with unlikely statin intolerance probably due to the non-attainment of recommended LDL-C targets.

It is not surprising that PAD patients represent a population of high cardiovascular risk, perhaps of the highest risk, when compared to patients with other atherosclerotic entities [7,20,21]. The intensive secondary prevention of major cardiovascular risk factors, particularly lipid lowering, is nowadays a well-established strategy, supported by the bulk of evidence from recent randomized trials of novel lipid-lowering agents [22,23,24], which has led to a further adjustment in the LDL-C target from below 1.8 mmol/L (70 mg/dL) to below 1.4 mmol/L (55 mg/dL) in the latest guidelines by the ESC/EAS [25]. Over decades, LDL-C target attainment in atherosclerotic patients has been a strenuous task related to low statin adherence rates, particularly in PAD patients where the adherence and lipid target attainment rates are even lower [7,8,9,26,27].

Any condition similar to statin intolerance, despite its low frequency in the population, further decreases statin adherence and target attainment rates, and renders a desired cardiovascular risk reduction nearly impossible and puts these patients in harm.

Statin intolerance is a severe condition, but to date, no agreement exists on the prevalence of true statin intolerance. Notably, most patients tolerate statins well, and the safety profile of statins is not significantly worse compared to a placebo in randomized controlled trials [1]. In contrast, greater numbers of patients experience statin-associated adverse effects in observational studies [28], which can, in part, be explained by the nocebo effect, a psychological phenomenon occurring when individuals with a preconceived negative expectation of an intervention report harm at a higher rate than anticipated [29]. Thus, individuals who have read about muscle-adverse effects related to statins are more likely to attribute any muscle pain to their prescribed statin. Recently, it was reported in a large statin outcomes trial that although muscle-related adverse effects did not differ between randomized groups during the initial blinded period, patients taking statins in the open-label extension phase of the study had a much higher rate of adverse effects [30]. In the present study, we cannot rule out a nocebo effect in the unlikely statin intolerance group. Furthermore, two-thirds of the reported muscle pains were mainly characterized as atypical and, not surprisingly, mostly in patients then categorized as having an unlikely statin intolerance. Their muscle-related symptoms were described by 94% of individuals in a way that would be characterized as atypical, i.e., asymmetric, distal, affecting smaller muscle groups. It should be emphasized that PAD patients often describe ischemic pain in the lower limbs while walking as a muscular cramp. Thus, some patients, either under the influence of a nocebo effect or not, might mistake their claudication pain for statin-related muscle pain.

In the present study, we observed a surprisingly low rate of SI in PAD patients, which became even lower after the classification as unlikely and probable/possible SI based on the SAMS-CI score [31]. The difficulty in identifying patients with likely statin intolerance might often be related to the highly heterogeneous clinical presentation of potential statin-related muscle symptoms. Muscle pain or aching, stiffness, tenderness, or cramp (often referred to as ‘myalgia’ [18]) attributed by patients to their statin use is usually symmetrical but may be localized and can be accompanied by muscle weakness. Any of these symptoms occur predominantly without an elevation of creatinine kinase (CK) (myalgia), but in some cases, CK elevation might be observed with or without symptoms (myopathy). In our population, CK levels were within a normal range in all patients, and none of the patients needed a biopsy. Muscle pain and weakness are usually symmetrical and proximal and generally affect large muscle groups including the thighs, buttocks, calves, and back muscles. Discomfort and weakness typically occur early (within 4–6 weeks after starting statin therapy [19]) but may still occur after many years of treatment. The onset of new symptoms may be associated with an increase in the statin dose or the initiation of a drug with interaction potential.

The large heterogeneity of muscle symptoms, either statin-induced or nocebo-related, surely contributes to the broad range of reported statin intolerance between 10 to 30% in patient registries [32,33,34] in contrast to randomized trials [9,35]. Furthermore, PAD patients might misinterpret ischemic muscle pain as statin-induced and vice versa, especially when they tend to experience a nocebo effect.

Altogether, distinguishing true SI from a variety of heterogeneous symptoms remains challenging. The implementation of the “statin myalgia clinical index score” [31] in the clinical routine offers clinicians valid support for the identification of true SI. In our study population, the application of the score retrospectively revealed that three-quarters of patients were unlikely to suffer from SI but had a higher mortality rate. It is likely that earlier identification may have reduced the high mortality in this population, since both the prescribed LLT and LDL-C lowering were less intense compared to the possible/probable SI group.

Another reason for the higher mortality in the unlikely SI group is that the non-adherence to statin therapy in our population over time was high, with a 40% drop in statin use and a change from high- to moderate-intensity statin therapy. However, LLT was surprisingly withdrawn in more patients in the unlikely statin intolerance group, leading to less reduced LDL-C levels. Taken together, the reluctance toward LLT in this group favored a reduced outcome observed by the higher mortality rate, although in principle, these patients could have received effective LLT. Unfortunately, the diagnosis of SI was maintained in patients’ files, exposing them to undertreatment.

## 5. Limitations

The results of this study represent a single-center experience, however within a large PAD population but with a small sample size to compare SI and a low number of CVE. As stated, some muscle symptoms might have been misinterpreted for ischemic pain during claudication. This might, however, also be true the other way around, thus the estimated number of unreported cases of true statin intolerance is likely to be higher.

In contrast, a potentially false positive diagnosis of SI might be caused by drug–drug- or food–drug interactions, which we did not test for due to missing information in this regard. Furthermore, some lab data were also not analyzed in the study, such as Vitamin D or the Thyroid stimulating hormone (TSH) since they are not generally in the lab routine of our department. Furthermore, we did not validate the SAMS-CL in our study population, i.e., by Cronbach’s alpha, and have not performed a multivariate analysis. 

## 6. Conclusions

The central finding of our study, namely a potentially lower true SI rate in PAD patients and probably related increased overall mortality in patients with unlikely SI, demonstrates the importance of early identification of patients with true statin intolerance and separating them from other patients likely not suffering from this condition. Especially nowadays, after the introduction of PCSK-9 inhibitors in daily clinical practice, powerful tools for proper and effective treatment and successful attainment of lipid targets are available and should be offered to patients in need, independently of the presence or absence of statin intolerance, in order to reduce the LDL-C burden and further decrease the cardiovascular risk.

## Figures and Tables

**Figure 1 jcm-11-06619-f001:**
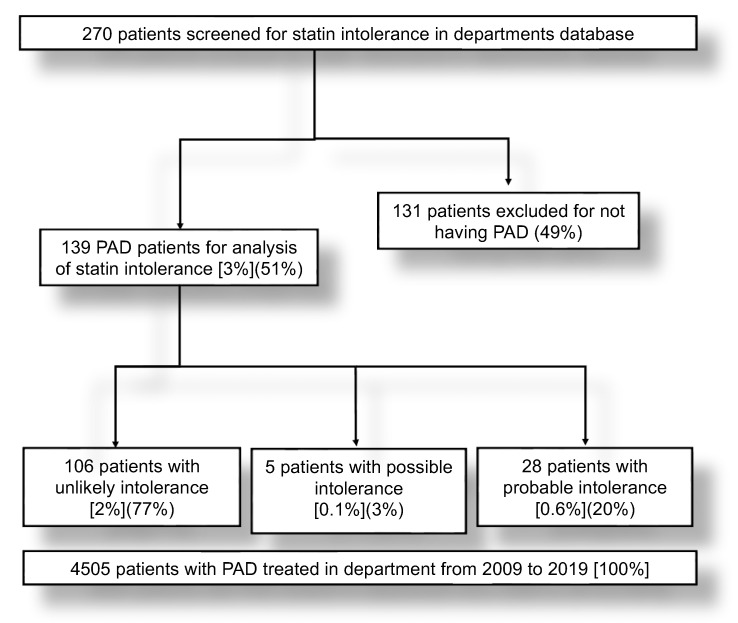
Flow chart of study population regarding identification of true from false statin intolerance. Of 270 patients, 139 patients were identified as having PAD. Proportion of patients regarding the study population analyzed is set in parentheses ( ). In regard to the total number of PAD patients treated at the department over the same period of time, the proportion is set in square brackets [ ].

**Figure 2 jcm-11-06619-f002:**
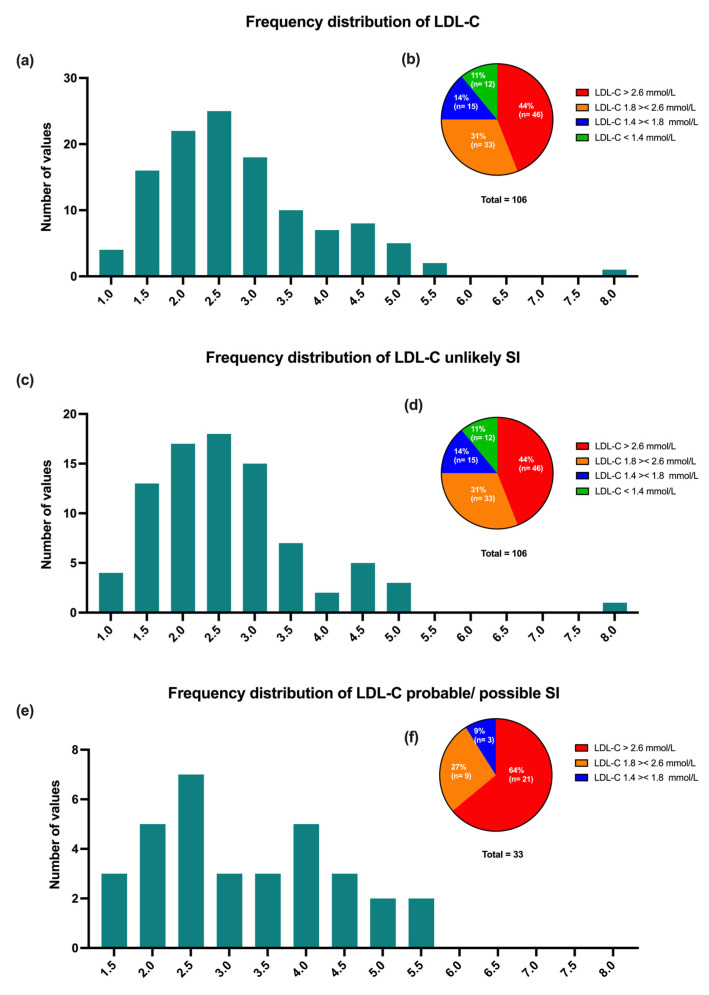
Patients with controlled LDL-C (**a**,**c**,**e**) and percentage of different LDL-C objective values (**b**,**d**,**f**) for different SI subgroups (all SI patients (**a**,**b**), unlikely statin intolerance (**c**,**d**), or probable/possible statin intolerance (**e**,**f**)). Data are presented as absolute numbers/percentages (*n*/N). LDL-C, low-density lipoprotein cholesterol.

**Figure 3 jcm-11-06619-f003:**
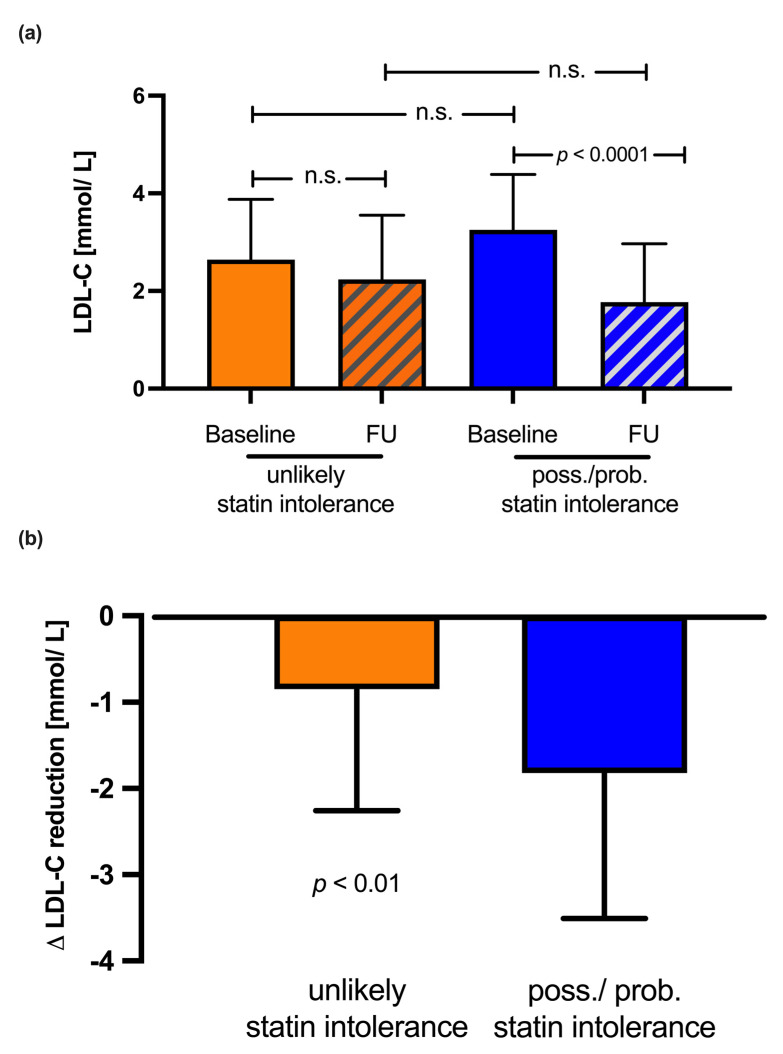
Change in LDL-C levels (**a**) as well as individual changes in LDL-C levels (**b**) over 62 months of follow-up. Data are presented as mean ± standard deviation. LDL-C, low-density lipoprotein cholesterol.

**Figure 4 jcm-11-06619-f004:**
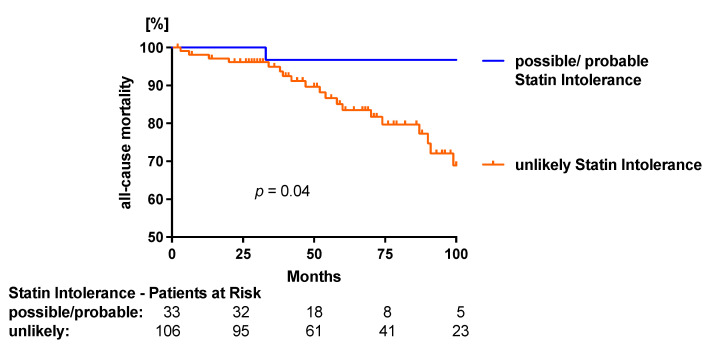
All-cause mortality of PAD patients with unlikely and probable/possible statin intolerance.

**Table 1 jcm-11-06619-t001:** Patient characteristics.

Baseline	All PAD Patients (*n* = 139)	Unlikely SI (*n* = 106)	Probable/Possible SI (*n* = 33)	*p*-Value
Age (years), mean ± SD	66.4 ± 9.9	68.1 ± 9.7	60.8 ± 8.7	**0.001**
Female Sex, *n* (%)	38 (27)	28 (26)	10 (30)	0.66
Current smoker, *n* (%)	43 (31)	30 (28)	13 (39)	0.28
Diabetes, *n* (%)	48 (35)	41 (39)	7 (21)	0.09
HbA1c [%]	6.19 ± 1.01	6.23 ± 10.7	6.07 ± 0.87	0.86
Hypertension, *n* (%)	122 (88)	91 (86)	31 (94)	0.31
Chronic Kidney disease (CKD), *n* (%)	84 (60)	73 (69)	18 (55)	0.14
Stage 2	54 (64)	44 (42)	12 (36)	0.69
Stage 3	25 (18)	21 (20)	5 (15)	0.62
Stage 4	5 (6)	8 (8)	1 (3)	0.67
Stage 5	0 (0)	0 (0)	0 (0)	
**Severity of PAD (Fontaine stage)**				
I, *n* (%)	49 (35)	40 (38)	9 (27)	0.3
II a, *n* (%)	27 (19)	18 (17)	9 (27)	0.21
II b, *n* (%)	43 (31)	32 (30)	11 (33)	0.83
III, *n* (%)	8 (6)	7 (7)	1 (3)	0.68
IV, *n* (%)	12 (9)	9 (8)	3 (9)	0.99
Creatinine, [µmol/L]	109.30 ± 87.40	118.10 ± 99.40	85.30 ± 28.15	**0.03**
eGFR [mL/min/1.73 m^2^]	67.87 ± 21.12	69.25 ± 22.14	77.52 ± 15.53	0.07
Total Cholesterol [mmol/L]	4.50 ± 1.23	4.35 ± 1.19	4.89 ± 1.28	**0.04**
HDL Cholesterol [mmol/L]	1.19 ± 0.36	1.22 ± 0.38	1.12 ± 0.29	0.22
Non-HDL [mmol/L]	3.28 ± 1.26	2.69 ± 1.55	3.77 ± 1.25	**<0.001**
LDL Cholesterol [mmol/L]	2.79 ± 1.18	2.62 ± 1.15	3.22 ± 1.14	**0.01**
Triglycerides [mmol/L]	2.11 ± 1.34	2.00 ± 1.35	2.39 ± 1.29	0.13
ALAT [U/L]	24.12 ± 17.40	22.70 ± 18.17	27.25 ± 15.10	0.06
ASAT [U/L]	25.21 ± 17.02	23.36 ± 17.26	29.31 ± 16.22	0.06
Creatine Kinase [U/L]	106.70 ± 109.10	102.40 ± 113.50	114.40 ± 97.78	0.30

Baseline characteristics for the general population (*n* = 139) and in regard to the probability of statin intolerance (SI). PAD: Peripheral artery disease.

**Table 2 jcm-11-06619-t002:** History of lipid-lowering therapy at baseline at the end of the observational period. Data are presented as absolute numbers/percentages (*n*/N) or means ± standard deviation (SD).

	Baseline	Final		Baseline			Final		
	All (*n*= 139)	All (*n*= 139)	*p*	Unlikely SI (*n* = 106)	Probable/ Possible SI (*n* = 33)	*p*	Unlikely SI (*n* = 106)	Probable/ Possible SI (*n* = 33)	*p*
**Statin therapy, *n* (%)**	119 (86)	63 (45)	**<0.0001**	87(82)	32 (97)	**0.04**	56 (53)	7 (21)	0.23
**Statin dose/d (mg ± SD)**	57.8 ± 34.9	50.8 ± 32.0	0.16	57.0 ± 32.9	58.7 ± 40.7	0.83	52.5 ± 32.9	37.1 ± 21.4	0.28
**Statin intensity, *n* (%)**									
High, *n* (%)	72 (61)	23 (37)	**0.003**	53 (61)	19 (59)	0.99	22 (39)	1 (14)	0.41
Moderate, *n* (%)	45 (38)	38 (60)	**0.005**	33 (38)	12 (38)	0.99	32 (57)	6 (86)	0.23
Low, *n* (%)	2 (2)	2 (3)	0.61	1 (1)	1 (3)	0.47	2 (4)	0 (0)	0.99
**Lipophil Statin, *n* (%)**	106 (58)	28 (44)	**<0.0001**	80 (87)	26 (69)	0.11	23 (41)	5 (71)	0.23
**Hydrophil Statin, *n* (%)**	76 (42)	35 (56)	0.07	54 (62)	22 (69)	0.16	33 (31)	2 (6)	**0.003**
**Ezetimib, *n* (%)**	26 (19)	35 (25)	0.25	19 (18)	5 (15)	0.80	26 (25)	9 (27)	0.82
**+Statin, *n* (%)**	24 (92)	18 (51)	**<0.001**	19 (100)	5 (100)	0.80	16 (62)	2 (22)	0.06
**+PCSK-9i, *n* (%)**	0 (0)	3 (9)	**<0.0001**	0 (0)	0 (0)		0(0)	3 (33)	**0.01**
**PCSK-9i, *n* (%)**	1 (1)	25 (18)	**<0.0001**	1 (1)	0 (0)	0.99	7 (7)	18 (55)	**<0.0001**
**+Statin, *n* (%)**	1 (100)	5 (20)	0.23	1 (100)	0 (0)	0.99	3 (43)	2 (11)	0.11
**+Ezetimib, *n* (%)**	0 (0)	3 (12)	0.99	0 (0)	0 (0)		0 (0)	3 (17)	0.53
**+Statin + Ezetimib, *n* (%)**	0 (0)	0 (0)		0 (0)	0 (0)		0 (0)	0 (0)	

## Data Availability

Data are stored in the PTA databank at Department of Angiology, Inselspital Bern, Switzerland.

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
