# Peer review of "False versus True Statin Intolerance in Patients with Peripheral Artery Disease"

_jcm, 2022, doi:10.3390/jcm11226619_

Round 1

Reviewer 1 Report

Dear Author(s),

Thank you for this interesting and clinically relevant manuscript. Findings regarding the real frequency of statin intolerance are highly important (as well as highlighting the consequences of misdiagnosing SI); especially since earlier distinguishing between true (probable) and unlikely SI may help reducing/preventing high mortality associated with unduly less intense LLT and LDL-C lowering in uSI patients.

Methodology as well as statistical analysis seems to be adequate and conclusions are supported by the results presented.

However, I have several suggestions/questions:

(I) Did you check for potential drug-drug interactions and food-drug interactions between groups? If not, maybe you should consider mentioning this in the limitation section.

(II) Please check numbers mentioned in Figure 1! In the figure you mentioned 271 and not 270 patients screened; also you mentioned 140 and not 139 patients for analysis of statin intolerance. Moreover, if I understood correctly, you screened 4505 PAD patients in total, why have you then excluded 131 patients for not having PAD (mentioned in Figure 1) - please explain.

(III) In the "Patients characteristics and identification of patients with possible or probable statin intolerance" section you used the term "more reduced kidney function", maybe it would be better to say "higher creatinine values", since there were no differences in terms of baseline CKD rate.

(IV) You mentioned that the majority of ppSI (88%) and uSI patients (47%) tried more than 1 statin. Please elaborate (especially regarding ppSI), since according to EAS definition you need to try at least 2 statins to diagnose with SI. I assume this is due to usage of statin myalgia clinical index score and associated classification?

(V) Please provide validation characteristics (e.g. Cronbach's alpha) for the statin myalgia clinical index score used. If not, consider mentioning in the limitation section.

(VI) Were there some other limitations/biases (everything that could influence the results presented) to disclose?

(VII) Do you maybe have sub-analysis data for all clinical outcomes for both groups separately - patients who started PCSK9 inhibitor therapy vs patients who did not?

Reviewer 2 Report

Statins are frequently prescribed for the treatment of cardio-vascular diseases and they belong to the most selling drugs worldwide. They are usually well tolerated but occasionally patients develop rhabdomyolysis and kidney failure. Here the authors report that among the 4,500 patients treated in their hospital with different types of statins for peripheral artery disease 139 cases (3 %) of statin intolerance (SI) were observed. Of those 76 % experienced a mild form of statin intolerance (uSI, uncertain SI) whereas 24 % developed more severe variants (ppSI, possible-probable SI). During treatment the statin use was significantly decreased in patients with both ppSI and uSI and at the end of the observation period, patients with ppSI more often received alternative therapy with PCSK9 inhibitors. From their data the authors concluded that statin intolerance is generally low (3 %). Moreover, inappropriate identification of statin intolerance, which frequently leads to dose reductions, is dangerous since it induces inadequate lipid lowering therapy, insufficient LDL cholesterol lowering, and increased mortality in patients with uncertain statin intolerance. Thus, clearly defined strategies for proper identification of statin intolerance are desperately needed in clinical praxis

The ms provides novel and interesting data, which are of immediate clinical relevance. It is clearly structured, well-written and even non-expert readers can easily follow the flow of information. The major limitations of this work are that the study was designed as single-centered study and that the number of possible/probable statin intolerance patients was relatively low (35). These limitations are openly discussed and thus, the ms can be published as it is without major modification. Unfortunately, the Abstract is full of abbreviations (SI, PAD, ppSI, uSI, SAMS-CI, LLT), which strongly impairs its legibility. For experts in the field the use of these abbreviations may not be problem but for average readers the excessive use of them may lead to confusions. Thus, I suggest minor textual revision of the abstract even if the word limit of the abstract is going to be exceeded.
